# Dynamic Focused Masking for Autoregressive Embodied Occupancy Prediction

**Yuan Sun[1]    Julio Contreras[1]    Jorge Ortiz[1]**
[1]Rutgers, The State University of New Jersey
New Brunswick, NJ, USA 08901
`ys820@soe.rutgers.edu`

## Abstract

Visual autoregressive modeling has recently demonstrated potential in image tasks by enabling coarse-to-fine, next-level prediction. Most indoor 3D occupancy prediction methods, however, continue to rely on dense voxel grids and convolution-heavy backbones, which incur high computational costs when applying such coarse-to-fine frameworks. In contrast, cost-efficient alternatives based on Gaussian representations—particularly in the context of multi-scale autoregression—remain underexplored. To bridge this gap, we propose DFGauss, a Dynamic Focused masking framework for multi-scale 3D Gaussian representation. Unlike conventional approaches that refine voxel volumes or 2D projections, DFGauss directly operates in the 3D Gaussian parameter space, progressively refining representations across resolutions under hierarchical supervision. Each finer-scale Gaussian is conditioned on its coarser-level counterpart, forming a scale-wise autoregressive process. To further enhance efficiency, we introduce an importance-guided refinement strategy that selectively propagates informative Gaussians across scales, enabling spatially adaptive detail modeling. Experiments on 3D occupancy benchmarks demonstrate that DFGauss achieves competitive performance, highlighting the promise of autoregressive modeling for scalable 3D occupancy prediction.

## 1 Introduction

With the accelerating progress in embodied intelligence and the deployment of active agents across domains such as robotics and autonomous navigation, spatial understanding has become a critical capability for intelligent systems [6, 40, 12]. To navigate indoor environments effectively, embodied agents must perform various perception tasks, among which occupancy prediction [38] plays a fundamental role in enabling agents to interpret and interact with complex real-world spaces. A key challenge in occupancy prediction lies in the trade-off between resolution and completeness: Small voxel sizes can cause holes and missing details, while larger voxels lead to over-smoothed, inaccurate geometry—issues that coarse-to-fine strategies address by progressively refining spatial resolution and structural fidelity [30]. Inspired by how humans perceive visual information—from global context to local detail in a hierarchical manner [29]—recent advances in autoregressive modeling [29, 28, 23, 20, 13] have shown great promise for addressing coarse-to-fine generation in 2D vision tasks, suggesting strong potential for enhancing 3D occupancy prediction.

While some efforts have attempted to extend autoregressive modeling to 3D volumetric representations using dense voxel grids [33, 1], these approaches often incur substantial computational overhead, limiting their scalability and generalizability. In contrast, efficient Gaussian-based representations offer a promising alternative, as they are both lightweight and capable of delivering strong performance. However, the autoregressive paradigm remains largely underexplored in the context of Gaussian-based methods, leaving a gap in effectively leveraging hierarchical modeling within computationally efficient 3D spatial frameworks.

39th Conference on Neural Information Processing Systems (NeurIPS 2025).

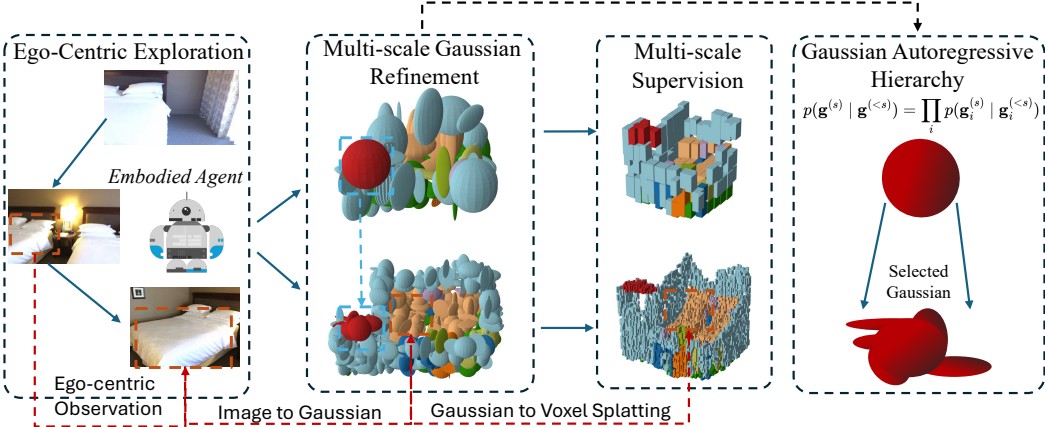

Figure 1: Overview of the proposed DFGauss framework. An embodied agent acquires egocentric RGB observations while navigating the environment, which are transformed into multi-scale Gaussian representations for autoregressive refinement. A selective masking strategy focuses computation on informative regions. In the illustrated example, the model identifies a bed (shown in peach) and ceiling (shown in dark red) through hierarchical Gaussian updates.

In this paper, we propose *DFGauss*, a Dynamic Focused masking framework for multi-scale 3D Gaussian representation (Figure 1). Our novelty lies in multi-scale Gaussian Splatting approach for 3D scene understanding that explicitly addresses the above mentioned issue through dynamic, attention-guided refinement. The core novelty of DFGauss lies in applying hierarchical autoregressive refinement directly to a sparse set of learnable 3D Gaussian primitives, avoiding reliance on dense volumetric or grid-based features. Unlike classical autoregressive models [29] that generate discrete tokens, DFGauss models spatial dependencies in a continuous space, where Gaussian parameters at each finer scale are predicted based on coarser-level outputs and corresponding encoder features. This mirrors the coarse-to-fine "next-scale prediction" strategy in Visual AutoRegressive (VAR) modeling [29], while diverging in its application to continuous Gaussian fields for structured 3D occupancy prediction. To improve efficiency, DFGauss incorporates a dynamic focused masking mechanism that selectively updates only the most informative Gaussians at each level, reducing computational cost without compromising reconstruction quality of this paradigm. Together, these components form a unified framework for efficient and expressive 3D spatial modeling, leading to the following contributions:

- A novel multi-scale autoregressive hierarchical 3D Gaussian Splatting framework that enhances 3D occupancy prediction for embodied agents.
- A coarse-to-fine supervision strategy that progressively refines Gaussian parameters across scales using multi-resolution labels.
- A dynamic focused masking mechanism that improves the efficiency of Gaussian refinement by selectively updating informative regions.
- Extensive experiments on multiple datasets demonstrating state-of-the-art performance in 3D occupancy prediction.

## 2 Related Work

### 2.1 3D Occupancy Prediction for Embodied Agents

Among various 3D perception tasks, occupancy prediction has emerged as a compact and expressive representation for modeling spatial semantics. While outdoor occupancy prediction has been extensively studied in the context of autonomous driving [33, 41, 32, 21, 9, 31, 4, 16], indoor environments remain relatively underexplored despite their critical role in embodied AI and robotics. MonoScene [3] proposes a voxel-based framework that infers occupancy from a single RGB image using a 2D-to-3D U-Net architecture with contextual priors. To enable real-time exploration, EmbodiedOcc [34] introduces a Gaussian-based memory refinement scheme. However, most existing

approaches still rely on dense 3D voxel grids [33, 19, 18], and current Gaussian-based methods remain in early stages of development. In particular, the rich features generated during Gaussian splatting have not been fully exploited. In this work, we propose a novel framework that refines sparse Gaussian features in a multi-scale autoregressive manner, combining the efficiency of Gaussian-based representations with enhanced accuracy in occupancy prediction.

## 2.2 Multi-scale Autoregression

Recent advances in visual autoregressive modeling (VAR) introduce a multi-scale next-resolution prediction strategy that amplifies supervision signals and enhances robustness, setting new benchmarks in 2D generation efficiency and scalability [29, 22, 11, 5]. Inspired by this, recent works have extended multi-scale modeling to 3D tasks such as occupancy prediction in autonomous driving. SurroundOcc [33], for instance, employs multi-scale 2D-3D attention to enable dense spatial reasoning and supervision. NOMAE [1] introduces a multi-scale self-supervised framework for LiDAR point clouds that focuses on localized occupancy reconstruction without modeling the full 3D volume. OctreeOcc [24] adopts an adaptive octree-based representation to support efficient and fine-grained occupancy prediction while reducing computational cost. Despite their differences, these methods rely on dense convolutional or transformer-based backbones. In contrast, the recent rise of Gaussian Splatting offers a cost-effective and compact alternative for 3D scene representation via continuous 3D Gaussian primitives. However, its integration with multi-scale optimization remains underexplored. We posit that coarse-to-fine autoregressive modeling in the scale space—where finer-scale representations are conditioned on coarser ones—provides an efficient and principled approach for structured refinement in sparse 3D representations.

## 2.3 Gaussian Splatting

3D Gaussian Splatting [14] has recently emerged as a compelling alternative to traditional volumetric and mesh-based rendering methods, offering real-time and high-fidelity radiance field rendering via anisotropic Gaussian primitives. Building on this foundation, a series of follow-up works aim to improve its efficiency and adaptability. Mip-Splatting [39] introduces a low-pass filtering mechanism to mitigate aliasing artifacts caused by the sampling-sensitive nature of splats. Multi-scale Gaussian Splatting [35] extends the framework to dynamic scenes by modeling object motion using MLPs. Other works focus on redundancy reduction: LightGaussian [7] and Compact3DGS [17] rank Gaussians by scale or opacity to prune uninformative primitives, significantly reducing memory and computation costs. Motivated by this line of research, we propose a dynamic focus masking mechanism that adaptively selects informative Gaussians across scales, enabling more efficient and scalable optimization in multi-scale 3D spaces, particularly for structured scene understanding, temporal consistency, and downstream prediction tasks in complex indoor environments with diverse spatial layouts and semantic variations common in embodied AI applications.

## 3 Method

Our model converts 2D image features into 3D Gaussian representations via cross-attention and sparse convolution. As illustrated in Figure 2, we introduce a hierarchical refinement framework that jointly optimizes multi-scale image features and Gaussian parameters across spatial resolutions. Given a monocular RGB image, a Hierarchical Feature Generation module extracts a multi-scale feature pyramid. To provide geometric cues, we incorporate a depth prediction network [36] to estimate depth maps at each level of the hierarchy. Each scale-specific feature map is then processed by a corresponding Gaussian Encoder, which predicts initial Gaussian parameters. Inspired by prior work [10, 34], we design a multi-scale Gaussian encoder that produces 3D Gaussians at different resolutions directly from the image features. These parameters are progressively refined by a Multi-Scale Gaussian Refinement module, where each level is conditioned on the output of the coarser scale via attention-based fusion, forming an autoregressive refinement process. For global Gaussian update, we adopt the same confidence-guided refinement strategy as in [34], selectively updating only the Gaussians corresponding to the original resolution. Finally, lightweight Gaussian-to-Voxel decoders apply 3D Gaussian splatting to project the refined Gaussians onto voxel grids, generating occupancy volumes at each resolution. Supervision is provided across all scales using multi-resolution 3D occupancy labels, facilitating effective learning of coarse-to-fine geometric structures. For training,

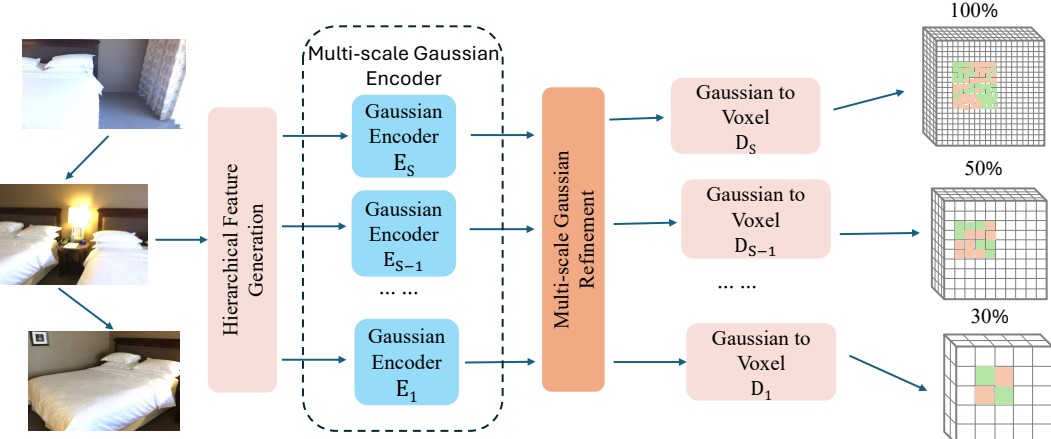

Figure 2: Overview of the DFGauss framework. Given an indoor monocular RGB image, a *Hierarchical Feature Generation* module extracts features at multiple granularity. These multi-scale features are encoded into Gaussian parameters by an *Multi-scale Gaussian Encoder*. The *Multi-Scale Gaussian Refinement* Module further optimizes these Gaussians across the hierarchy. Finally, a *Gaussian-to-Voxel* decoder predicts 3D occupancy from the multi-scale features. Supervision is provided by 3D occupancy labels at multiple resolutions.

we adopt the loss formulation from [10], combining Focal Loss, Lovász Loss, Semantic-Scale Loss, and Geometric-Scale Loss to jointly address class imbalance, boundary precision, and multi-scale consistency. In summary, our framework establishes a fully image-based Gaussian representation pipeline for monocular 3D occupancy prediction across multiple spatial granularities.

### 3.1 Multi-Scale Gaussian Encoder

Inspired by recent works [10, 34], we propose a Gaussian encoder that operates at each hierarchical scale of the image feature pyramid. Given image features at a specific scale $s$, the encoder first initializes a sparse set of 3D Gaussian primitives within camera frustum. Each Gaussian is parameterized by a tuple $\mathbf{g}_i^{(s)} = \{\mu_i, \boldsymbol{\lambda}_i, q_i, o_i, \mathbf{l}_i\}$, where $\mu_i \in \mathbb{R}^3$ is the mean position, $\boldsymbol{\lambda}_i \in \mathbb{R}^3$ are scale factors, $q_i \in \mathbb{R}^4$ is rotation quaternion, $o_i \in \mathbb{R}$ is the opacity, and $\mathbf{l}_i \in \mathbb{R}^C$ are semantic logits over $C$ classes.

To incorporate visual cues, each Gaussian is lifted into a high-dimensional feature space via a feature alignment module, which integrates local image descriptors with geometric priors from a depth-aware branch. These features are then refined through a combination of self-attention (among Gaussians) and cross-attention (with multi-scale image features), enabling context-aware adaptation to the observed scene geometry.

To extend this to a multi-scale design, we construct a hierarchy of Gaussian encoders $\{E^{(s)}\}_{s=1}^S$ operating over $S$ scales of image features $\{\mathbf{F}^{(s)}\}_{s=1}^S$, ordered from coarse ($s = 1$) to fine ($s = S$). Each encoder $E^{(s)}$ predicts a set of Gaussians $\mathbf{G}^{(s)}$ from its input features. Formally, we define:

$$\mathbf{G}^{(s)} = E^{(s)}(\mathbf{F}^{(s)}, D^{(s)}), \quad \mathbf{G}^{(s)} = \left\{ \mathbf{g}_i^{(s)} \right\}_{i=1}^{N_s}, \quad \forall s \in \{1, \dots, S\}, \tag{1}$$

where $D^{(s)}$ is the predicted depth map at scale $s$, used for geometric alignment during Gaussian initialization. Each $\mathbf{g}_i^{(s)}$ is defined in 3D space, enabling sparse yet expressive representation.

This multi-scale Gaussian encoding framework allows each level to specialize in a different spatial granularity, facilitating coarse-to-fine 3D understanding and efficient downstream refinement.

### 3.2 Hierarchical Multi-Scale Refinement in Gaussian Parameter Space

We propose to perform hierarchical refinement directly on the parameters of 3D Gaussians. This representation offers three key advantages: (*i*) compactness—Gaussian primitives provide a sparse,

continuous encoding of the scene without voxel quantization artifacts; (*ii*) expressiveness—each Gaussian captures both spatial location and geometric shape via learnable scale and orientation; and (*iii*) hierarchical alignment—different scales of Gaussians naturally correspond to varying levels of scene abstraction. These properties make the Gaussian parameter space a compelling domain for multi-resolution refinement, allowing the network to progressively increase spatial fidelity while preserving semantic structure. The core innovation of our approach lies in shifting the multi-scale refinement paradigm from dense voxel-based feature maps to a sparse, continuous set of learnable 3D Gaussians. By operating directly in the Gaussian parameter space, our method achieves finer geometric detail, efficient memory scaling, and hierarchical abstraction across spatial resolutions.

We explicitly formulate this process as an autoregressive refinement hierarchy for 3D scene representation, where the Gaussian set at each scale $s$ is conditioned on all coarser levels $(1{:}s{-}1)$. Specifically, we model the hierarchical dependency as:

$$p\Big(\mathbf{G}^{(S)} \,\Big|\, \mathbf{G}^{(1:S-1)}, \hat{\mathbf{G}}^{(1:S)}\Big) = \prod_{s=1}^{S} p\Big(\mathbf{G}^{(s)} \,\Big|\, \mathbf{G}^{(1:s-1)}, \hat{\mathbf{G}}^{(s)}\Big). \tag{2}$$

Here, $\hat{\mathbf{G}}^{(s)}$ denotes the initial Gaussian set predicted at scale $s$, and $\mathbf{G}^{(s)}$ represents its refined version obtained by updating $\hat{\mathbf{G}}^{(s)}$ with residual corrections guided by $\mathbf{G}^{(1:s-1)}$. This autoregressive structure enables structured, scale-wise propagation of geometric and semantic information.

Given the coarse-to-fine Gaussian sets $\mathbf{G}^{(1:s-1)}$ and the initial prediction $\hat{\mathbf{G}}^{(s)}$, we compute residual updates via a cross-attention fusion block:

$$\Delta\mathbf{G}^{(s)} = f_{\text{attn}}^{(s)}\Big(\mathbf{G}^{(1:s-1)}, \hat{\mathbf{G}}^{(s)}\Big), \tag{3}$$

where $f_{\text{attn}}^{(s)}$ denotes a learned attention-based MLP that integrates coarse geometric priors with initial scale-$s$ predictions for residual parameter refinement.

The refined Gaussian parameters at scale $s$ are obtained by updating the initial predictions with learned residuals. For position, scale, opacity, and semantic logits, we have

$$\{\mu_i^{(s)}, \boldsymbol{\lambda}_i^{(s)}, o_i^{(s)}, \mathbf{l}_i^{(s)}\} = \{\hat{\mu}_i^{(s)} + \Delta\mu_i^{(s)}, \ \hat{\boldsymbol{\lambda}}_i^{(s)} + \Delta\boldsymbol{\lambda}_i^{(s)}, \ \hat{o}_i^{(s)} + \Delta o_i^{(s)}, \ \hat{\mathbf{l}}_i^{(s)} + \Delta\mathbf{l}_i^{(s)}\}. \tag{4}$$

The rotation is updated separately by refining the initial quaternion prediction $\hat{\mathbf{q}}_i^{(s)}$ with a learned delta quaternion $\Delta\mathbf{q}_i^{(s)}$:

$$\mathbf{q}_i^{(s)} = \text{Normalize}\Big(\hat{\mathbf{q}}_i^{(s)} \otimes \Delta\mathbf{q}_i^{(s)}\Big), \tag{5}$$

where $\otimes$ denotes quaternion multiplication and $\texttt{Normalize}$ enforces the unit-norm constraint. Here, $\hat{\mathbf{q}}_i^{(s)}$ is the initial rotation quaternion of the $i$-th Gaussian at scale $s$, $\Delta\mathbf{q}_i^{(s)}$ is the learned quaternion update, and $\mathbf{q}_i^{(s)}$ is the final normalized quaternion rotation after refinement.

This formulation enables structured multi-scale refinement while maintaining geometric and semantic coherence across scales and resolution levels.

### 3.3  Selective Gaussian Refinement Mask

To further enhance optimization efficiency, we introduce a *Selective Gaussian Refinement Mask*. The core idea is to leverage coarse-scale importance cues to identify spatially relevant regions and selectively refine only a sparse subset of fine-scale Gaussian parameters (Figure 3). By concentrating computational resources on informative areas, this mask improves both efficiency and scalability for training high-resolution 3D Gaussian fields.

Let $\mathbf{G}^{(s-1)} \in \mathbb{R}^{N_{s-1} \times D}$ and $\mathbf{G}^{(s)} \in \mathbb{R}^{N_s \times D}$ denote the Gaussian parameters at scales $s{-}1$ and $s$, respectively. Rather than refining fine-scale Gaussians based on all $N_{s-1}$ coarse-scale anchors, we identify and propagate information from a smaller, semantically relevant subset.

**Percentile-Based Importance Selection.** Each coarse anchor $\mathbf{g}_i^{(s-1)}$ is assigned a scalar importance score $\pi_i^{(s-1)}$ via a learnable scoring function:

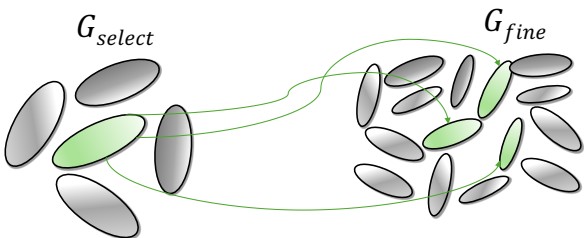

Figure 3: Illustration of the Selective Gaussian Refinement Mask. Coarse-scale Gaussians with high importance scores are first selected (left) and then used to guide the refinement of fine-scale Gaussians (right) through cross-scale attention.

$$\pi_i^{(s-1)} = \text{ScoreNet}(\mathbf{g}_i^{(s-1)}), \quad \pi_i^{(s-1)} \in \mathbb{R}. \tag{6}$$

We compute importance scores $\pi_i^{(s-1)}$ for each coarse-scale anchor using a lightweight MLP-based scoring function. We then retain the top-$\rho\%$ of anchors ranked by $\pi_i^{(s-1)}$, forming the index set:

$$\mathcal{I}_{\text{select}} = \text{Top}_\rho\left(\{\pi_i^{(s-1)}\}_{i=1}^{N_{s-1}}\right), \quad \mathbf{G}_{\text{select}}^{(s-1)} = \left\{\mathbf{g}_i^{(s-1)} \mid i \in \mathcal{I}_{\text{select}}\right\}. \tag{7}$$

Here, we apply a soft index set corresponding to the top-$\rho\%$ of Gaussians [26, 2]. Let $\mathcal{I}_{\text{select}} \in \mathbb{R}^{B \times N \times K}$ be a soft index set, where $B$ is the batch size, $N$ is the number of Gaussians, and $K$ is the top-$\rho\%$ number of Gaussians. Each slice over the last dimension defines a distribution over the $N$ elements, indicating their soft selection probability of being in the top-$\rho\%$. This formulation identifies regions requiring further refinement, as $\mathbf{G}_{\text{select}} \in \mathbb{R}^{B \times K \times D}$ is selected from the original matrix $\mathbf{G} \in \mathbb{R}^{B \times N \times D}$, where $D$ is the feature dimension. During inference, we replace this relaxation with a discrete $\text{Top}_k$ operator for both memory and computational efficiency.

This subset serves as a coarse-to-fine attention mask that identifies regions requiring further refinement in both geometry and semantics.

**Cross-Scale Attention for Refinement.** Let $\mathbf{G}_{\text{fine}}^{(s)} \in \mathbb{R}^{B \times M \times D}$ and $\mathbf{G}_{\text{select}}^{(s-1)} \in \mathbb{R}^{B \times K \times D}$ denote the fine-scale and selected coarse-scale Gaussian sets, respectively, where $B$ is the batch size, $D$ is the feature dimension, and $M, K$ are the number of queried and key-value Gaussians per sample.

We apply a cross-scale attention mechanism as follows:

$$\mathbf{G}_{\text{fine}}^{(s)} = \hat{\mathbf{G}}_{\text{fine}}^{(s)} \oplus \text{softmax}\left(\frac{\hat{\mathbf{G}}_{\text{fine}}^{(s)}(\mathbf{G}_{\text{select}}^{(s-1)})^\top}{\sqrt{D}}\right)\mathbf{G}_{\text{select}}^{(s-1)}. \tag{8}$$

Here, $\oplus$ denotes the residual update operator in the Gaussian parameter space: quaternion components are composed, while position, scale, opacity, and semantic logits are updated through element-wise addition, consistent with Eqs. (4)–(5). The resulting $\mathbf{G}_{\text{fine}}^{(s)}$ retains the same shape $(B, M, D)$.

## 4 Experiment

We conduct experiments on three indoor occupancy prediction benchmarks: Occ-ScanNet [38], EmbodiedOcc-ScanNet [34], and their respective smaller variants, Occ-ScanNet-mini [34] and EmbodiedOcc-ScanNet-mini [34]. Occ-ScanNet and its mini version provide monocular RGB inputs paired with voxel-level semantic labels within a frustum-aligned 3D space, supporting per-frame local occupancy prediction in a static setting. In contrast, EmbodiedOcc-ScanNet introduces a sequential and embodied formulation, where temporally continuous monocular observations enable iterative refinement of global scene understanding. Each frame is labeled with a local occupancy volume projected from a globally consistent ground-truth space, allowing both frame-wise supervision and memory-based global prediction. Following standard protocols [38, 34, 33], we report Scene

Table 1: **Local Prediction (Single-View) Results on the Occ-ScanNet dataset.**

| Method | Input | IoU | ceiling | floor | wall | window | chair | bed | sofa | table | tvs | furniture | objects | mIoU |
|---|---|---|---|---|---|---|---|---|---|---|---|---|---|---|
| MonoScene [3] | $x^{\text{rgb}}$ | 41.60 | 15.17 | 44.71 | 22.41 | 12.55 | 26.11 | 27.03 | 35.91 | 28.32 | 6.57 | 32.16 | 19.84 | 24.62 |
| ISO [38] | $x^{\text{rgb}}$ | 42.16 | 19.88 | 41.88 | 22.37 | 16.98 | 29.09 | 42.43 | 42.00 | 29.60 | 10.62 | 36.36 | 24.61 | 28.71 |
| EmbodiedOcc [34] | $x^{\text{rgb}}$ | 53.95 | 40.90 | 50.80 | 41.90 | 33.00 | 41.20 | 55.20 | 61.90 | 43.80 | 35.40 | 53.50 | 42.90 | 45.48 |
| Ours | $x^{\text{rgb}}$ | **55.28** | **42.23** | **52.95** | **43.23** | **34.20** | **43.20** | **56.73** | **63.81** | **45.66** | **35.92** | **55.23** | **44.33** | **47.03** |

Table 2: **Local Prediction Results on the Occ-ScanNet-mini dataset.**

| Method | Input | IoU | ceiling | floor | wall | window | chair | bed | sofa | table | tvs | furniture | objects | mIoU |
|---|---|---|---|---|---|---|---|---|---|---|---|---|---|---|
| MonoScene [3] | $x^{\text{rgb}}$ | 41.90 | 17.00 | 46.20 | 23.90 | 12.70 | 27.00 | 29.10 | 34.80 | 29.10 | 9.70 | 34.50 | 20.40 | 25.90 |
| ISO [38] | $x^{\text{rgb}}$ | 42.90 | 21.10 | 42.70 | 24.60 | 15.10 | 30.80 | 41.00 | 43.30 | 32.20 | 12.10 | 35.90 | 25.10 | 29.40 |
| EmbodiedOcc [34] | $x^{\text{rgb}}$ | 53.80 | 29.10 | 48.70 | **42.30** | 38.70 | 42.00 | 62.70 | 60.60 | 48.20 | 33.80 | **58.00** | 46.50 | 46.40 |
| Ours | $x^{\text{rgb}}$ | **54.28** | **29.25** | **48.73** | 38.70 | **39.28** | **42.33** | **64.88** | **62.56** | **49.73** | **37.13** | 57.29 | **48.07** | **47.08** |

Completion Intersection-over-Union (IoU) and mean Intersection-over-Union (mIoU) across semantic classes. Local metrics are computed within the camera frustum of each frame, while global metrics evaluate performance over the union of explored regions, capturing the model's ability to maintain spatial consistency over time. While our primary focus is on indoor occupancy prediction, we further assess the generalization capability of our approach on outdoor scenarios using publicly available datasets. These additional experiments underscore the broader applicability of our method. Experimental details and additional evaluation results are provided in Appendix.

## 4.1 Main Results

**Local Prediction (Single-View) Results.** In the single-view setting, DFGauss consistently outperforms prior baselines across datasets of varying scales. On the Occ-ScanNet benchmark (Table 1), our method achieves an IoU of 55.28 and mIoU of 47.03, surpassing the Gaussian-based baseline [34] by 1.33% and 1.55% respectively. This highlights the advantage of our multi-scale autoregressive Gaussian refinement. On the smaller-scale Occ-ScanNet-mini dataset (Table 2), DFGauss further improves performance with an IoU of 54.28 and mIoU of 47.08, outperforming all voxel-based and Gaussian-based baselines. These consistent gains across both full and mini variants demonstrate the robustness and scalability of our approach for local occupancy prediction. These results demonstrate that the hierarchical multi-scale design of DFGauss enables robust generalization under limited training data and across diverse model architectures and input modalities.

**Global Prediction (Continuous-View) Results.** In the continuous-view setting, DFGauss also outperforms the Gaussian-based baseline (Table 3). The proposed multi-scale refinement enhances scene detail modeling and facilitates better alignment between the current view and previously observed frames. On the EmbodiedOcc and EmbodiedOcc-mini datasets, DFGauss surpasses the single-scale Gaussian baseline(Table 4), achieving a 1.36% improvement in embodied mIoU and a 1.44% gain in mIoU on the mini subset. These results highlight the robustness of DFGauss and the effectiveness of hierarchical multi-scale regression in capturing fine-grained temporal and spatial context through dynamic, agent-centric environments.

## 4.2 Ablation Study

**Component-Wise Ablation.** In the component-wise ablation study (Table 5), DFGauss achieves the highest performance when both the multi-scale regression module and the dynamic masking strategy are integrated. Compared to the vanilla model, introducing the multi-scale regression alone improves

Table 3: **Global Prediction (Continuous-View) Results on the EmbodiedOcc-ScanNet Dataset.**

| Method | Input | IoU | ceiling | floor | wall | window | chair | bed | sofa | table | tvs | furniture | objects | mIoU |
|---|---|---|---|---|---|---|---|---|---|---|---|---|---|---|
| SplicingOcc | $x^{rgb}$ | 49.01 | **31.60** | 38.80 | 35.50 | 36.30 | 47.10 | 54.50 | 57.20 | 34.40 | 32.50 | 51.20 | 29.10 | 40.74 |
| EmbodiedOcc[34] | $x^{rgb}$ | 51.52 | 22.70 | 44.60 | **37.40** | 39.00 | 50.10 | 56.70 | 59.70 | 35.40 | 38.40 | 52.00 | 32.90 | 42.53 |
| Ours | $x^{rgb}$ | **53.80** | 26.35 | **46.22** | 36.73 | **39.22** | **50.37** | **57.41** | **60.21** | **37.50** | **42.22** | **53.20** | **33.25** | **43.89** |

Table 4: **Global Prediction Results on the EmbodiedOcc-ScanNet-mini dataset.**

| Method | Input | IoU | ceiling | floor | wall | window | chair | bed | sofa | table | tvs | furniture | objects | mIoU |
|---|---|---|---|---|---|---|---|---|---|---|---|---|---|---|
| SplicingOcc | $x^{rgb}$ | 48.80 | **29.00** | 37.60 | 37.30 | 26.80 | 44.50 | 66.00 | 52.70 | 40.80 | 36.60 | 54.50 | 27.90 | 41.20 |
| EmbodiedOcc [34] | $x^{rgb}$ | 50.70 | 21.50 | 44.50 | 38.30 | 27.90 | 46.90 | 64.70 | 55.30 | 42.70 | 35.80 | 52.50 | 27.50 | 41.60 |
| Ours | $x^{rgb}$ | **52.32** | 22.73 | **44.80** | **38.70** | **30.21** | **47.12** | 65.10 | **55.62** | **43.21** | **37.83** | **55.24** | **32.83** | **43.04** |

IoU by 1.85% and mIoU by 0.78%. When the dynamic mask is further applied, the gains increase to 2.71% in IoU and 1.13% in mIoU. These results indicate that the multi-scale hierarchy effectively refines the details of occupancy prediction, while the dynamic mask focuses computation on regions requiring the most refinement.

**Mask Ratio Ablation.** In the mask ratio ablation study (Table 6), we observe that increasing the mask ratio up to 50% consistently improves both local and global performance metrics, indicating that denser refinement improves representational quality. However, beyond this point, further increases do not yield consistent gains and may slightly degrade performance, likely due to overly sparse Gaussian selections limiting effective refinement. Additionally, lower mask ratios lead to reductions in memory usage, reflecting the efficiency benefits of selectively refining only the most informative regions.

**Hierarchy Level Ablation.** As shown in the hierarchy level ablation (Table 7), performance steadily improves with increasing refinement depth up to level 4, after which the gains plateau or slightly decline under a fixed masking ratio. Notably, the latency increases gradually—from 165 ms to 287 ms—as the refinement depth grows, indicating that deeper hierarchies incur only moderate computational overhead. This suggests that our Gaussian-based framework enables scalable multi-scale refinement while maintaining reasonable inference efficiency.

Table 5: Ablation on the Components

| Multi-Scale | Mask | Local Metric | | Global Metric | |
|---|---|---|---|---|---|
| | | IoU | mIoU | IoU | mIoU |
| - | - | 52.33 | 46.20 | 51.33 | 42.46 |
| ✓ | - | 54.32 | 46.65 | 53.04 | 43.21 |
| ✓ | ✓ | **55.28** | **47.03** | **53.80** | **43.89** |

Table 6: Effect of mask ratio on inference memory and performance

| Ratio% | Memory | Local Metric | | Global Metric | |
|---|---|---|---|---|---|
| | | IoU | mIoU | IoU | mIoU |
| 80 | 5328 M | 54.03 | 46.12 | **53.83** | 43.47 |
| 70 | 5524 M | **55.82** | 46.54 | 53.72 | 43.63 |
| 60 | 5732 M | 55.28 | **47.03** | 53.80 | **43.89** |
| 50 | 5998 M | 55.68 | 46.93 | 53.76 | 43.58 |
| 40 | 6179 M | 54.84 | 46.73 | 53.42 | 43.21 |

## 4.3 Qualitative Results

Figure 4 presents qualitative results on the Occ-ScanNet dataset. Compared to the Gaussian-based baseline without autoregressive hierarchy, DFGauss captures more fine-grained structures in the target space, demonstrating the benefit of multi-scale refinement in improving spatial detail.

Table 7: Ablation on hierarchical depth; depth 1 is the baseline without refinement.

| Depth | Latency (ms) | Local Metric | | Global Metric | |
|---|---|---|---|---|---|
| | | IoU | mIoU | IoU | mIoU |
| 1 | 165 | 52.33 | 46.20 | 51.33 | 42.46 |
| 2 | 193 | 54.67 | 46.88 | 52.25 | 42.95 |
| 3 | 232 | 55.28 | **47.03** | **53.8** | **43.89** |
| 4 | 268 | **55.89** | 46.78 | 53.65 | 43.72 |
| 5 | 287 | 55.13 | 46.83 | 53.83 | 43.62 |

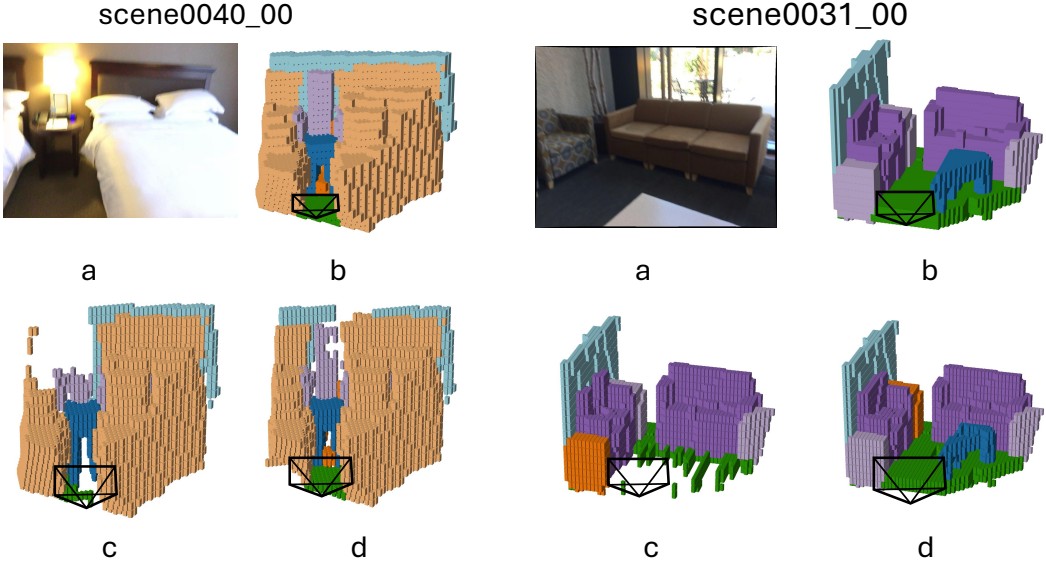

Figure 4: Qualitative results on Occ-ScanNet: (a) Input, (b) GT, (c) EmbodiedOcc, (d) DFGauss.

## 5 Limitation and Future Work

In this paper, we primarily focus on a Gaussian-based model architecture for monocular indoor occupancy prediction. However, extending this framework to multi-view indoor perception [34] remains an open direction and is not explored in this work. Additionally, the proposed mechanism could be applied to a broader range of point cloud tasks, such as panoptic segmentation [25, 37], semantic segmentation [27], occupancy prediction [38], and 3D scene completion [15, 8]. In future work, we plan to extend our experiments to these 3D scene understanding tasks to further investigate the potential of multi-scale Gaussian representations across diverse domains.

## 6 Conclusion

We present DFGauss, a multi-scale autoregressive Gaussian framework for 3D occupancy prediction. Unlike traditional approaches that rely on dense volumetric multi-scale regression, our method focuses on learning in the Gaussian parameter space across scales—a sparse and efficient representation that remains underexplored. To further enhance both accuracy and efficiency, we introduce a selective, dynamically focused refinement mask that prioritizes informative regions during hierarchical refinement. Extensive experiments demonstrate that our approach improves occupancy prediction performance while also reducing computational overhead. We hope that this framework inspires future research on efficient and expressive 3D scene understanding.

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
