# OpenReview forum: "Dynamic Focused Masking for Autoregressive Embodied Occupancy Prediction"
_NeurIPS.cc/2025/Conference — NeurIPS 2025 poster_

### Official Review · Reviewer_SDTU · 2025-06-10

**Clarity:** 3
**Significance:** 1
**Originality:** 2
**Rating:** 4
**Confidence:** 4

**Summary:**

This paper presents an autoregressive framework for efficient 3D occupancy prediction using multi-scale Gaussian representations. By progressively refining sparse 3D Gaussians through dynamic masking, the proposed DFGauss method aims to improve prediction accuracy while reducing computational cost, demonstrating promising results on standard benchmarks.

**Questions:**

* Provide a more detailed justification for the choice of occupancy grids in embodied scenarios, including a discussion of their benefits over depth-based or direct 3D Gaussian approaches. It would be better if the authors could show some real applications of the occupancy in embodied scenarios.
* Include quantitative comparisons of inference speed and memory usage with other leading methods.
* Clarify any methodological distinctions between occupancy prediction for autonomous driving and embodied perception tasks.

If these points are thoroughly addressed, I would be inclined to raise my rating.

**Ethical Concerns:**

["NO or VERY MINOR ethics concerns only"]

**Final Justification:**

The author has addressed my problems during rebuttal and discussion, especially about the motivation for embodied occupancy.

**Limitations:**

Yes

**Paper Formatting Concerns:**

No paper formatting concerns.

**Quality:**

2

**Strengths And Weaknesses:**

Strengths
* **Methodological Design:** The DFGauss framework introduces a hierarchical, autoregressive process for 3D Gaussian representations, enabling coarse-to-fine occupancy prediction. The dynamic masking strategy selectively refines only the most informative Gaussians, which leads to more efficient training and inference.
* **Clarity of Writing:** The paper is well-written, clearly articulating both the problem and the proposed methodology.

Weaknesses
* **Limited Novelty:** The use of self-attention and cross-attention mechanisms for generating 3D Gaussian representations has already been widely explored in the 3D AIGC field [1, 2]. Furthermore, as acknowledged by the authors, coarse-to-fine pipelines are standard practice in occupancy prediction, limiting the novelty of the approach.
* **Inference Speed and Memory Consumption:** While the coarse-to-fine pipeline can boost accuracy, it may also result in slower inference, as noted in Table 7 in the paper. It would strengthen the work to include a comprehensive comparison of inference speed and memory consumption with other methods.
* **Motivation for Embodied Occupancy:** The motivation for embodied occupancy prediction is not entirely convincing. In autonomous driving, occupancy grids are widely used because they provide basic geometric reconstruction from just visual input, with a typical voxel resolution of 200×200×16, which suffices for outdoor scenes. For embodied scenarios—***where finer geometric detail is crucial for manipulation and other tasks***—the paper adopts a lower voxel resolution (60×60×36), which may be inadequate. It remains unclear what advantages embodied occupancy offers over alternative perception methods such as direct depth estimation or 3DGS reconstruction. Additionally, the paper does not sufficiently discuss methodological differences between occupancy prediction for autonomous driving and embodied perception. Finally, it remains unclear why not use the reconstructed 3D Gaussian representations directly, as voxelization may discard valuable geometric information.

[1] Chen et al., *3DTopia-XL: Scaling High-quality 3D Asset Generation via Primitive Diffusion*, CVPR 2025.

[2] Zhang et al., *GeoLRM: Geometry-Aware Large Reconstruction Model for High-Quality 3D Gaussian Generation*, NeurIPS 2024.

---

> ### Author Rebuttal · Authors · 2025-07-30
>
> Dear Reviewer SDTU,
>
> Thank you for acknowledging the clarity of our writing and the design of our framework. Below are our detailed responses to your feedback.
>
> ## **Q1 Detailed justification for the choice of occupancy grids.(Weakness 3 Motivation)**
>
> The **choice of occupancy grids** is primarily determined by the baseline model and dataset specifications. It depends on the resolution provided by the original dataset, and all baselines adopt the same grid size to ensure fair comparison.
>
> The key **advantage** of monocular semantic occupancy prediction is that it can implement **Occlusion reasoning**[7] to understand and interpret scenarios where objects are partially or fully obscured by others[9][10]. Additionally, its cost efficiency and rich visual information have garnered significant attention in the robotics community[7]. It enables the inference of complete 3D geometry and semantic understanding of scenes using only 2D images[8].
>
> Traditional monocular **depth estimation** aims to infer the distance from the camera center to the 3D **surface** point corresponding to each pixel in a single RGB image, typically by predicting a depth map. However, depth maps only predict the nearest occupied point in each optical ray and are unable to recover the occluded parts of the 3D scene[18]. This depth information is often used as an **intermediate** representation for downstream tasks such as 3D occupancy prediction, as seen in our baselines in EmbodiedOcc (ICCV 2025) and ISO (ECCV 2024).
>
> Dense representation is computationally expensive in robotics[13]. Occupancy prediction models often employ sparse formats, where only occupied voxels (or cells) are explicitly stored, reducing computational overhead and memory consumption[14][15]. For embodied perception, such as robotic navigation, a compact, discrete, memory-efficient, and uniform representation is essential[16]. In contrast, **Direct 3D Gaussian** approaches **can not naturally** provide these characteristics. Gaussian-based method can **contribute** to occupancy estimation in scale-aware training and faster rendering in the voxel space[11]. Importantly, predicting 3D semantic occupancy from Gaussian representations is **different** from 3D Gaussian splatting[12]. Thus, Gaussian-based occupancy prediction is a well-suited task format that can meet the above requirement.
>
> Occupancy grids are a common choice in robotics, primarily due to their explicit free-space definition and their ability to handle unobserved areas through inference and completion [6]. There is a wide range of **applications for occupancy prediction** in embodied scenarios, such as robotic navigation [1], planning [2], obstacle detection [5], and mapping [3,4].
>
> ## **Q2 Quantitative comparisons...**
>
> | Method                    | Latency (ms) | Memory (MB) |
> |---------------------------|--------------|-------------|
> | ISO (ECCV 2024 )          |  650         | 10214       |
> | EmbodiedOcc (ICCV 2025)   | 114          | 3161        |
> | Ours                      | 232          | 5732        |
>
> As shown in the table, all methods are evaluated under the same setup. Compared to traditional convolution-based architectures such as ISO and Gaussian-based models like EmbodiedOcc, our method achieves a favorable balance between efficiency and accuracy.
>
> ## **Q3 Methodological distinctions**
>
> Indoor scenes typically exhibit more **diverse** and complex changes, such as variations in illumination, geometry, and appearance due to human interaction, making them harder to predict compared to outdoor scenes, which often have cyclic changes like day-night and seasonal variations[17]. In contrast, outdoor driving environments typically involve large-scale, **structured** spaces such as roads, intersections, and lanes, where objects like vehicles and pedestrians follow predictable spatial patterns[6]. Thus, feature refinement in the indoor scenario is important, while scalable training[11] is crucial for outdoor driving scenarios.
>
> From a methodological perspective, indoor scenes benefit more from voxel-based depth fusion due to the critical role of height information and complex object layouts in enclosed environments. In contrast, mainstream BEV-based depth fusion, which performs well in outdoor driving scenes, is less effective when directly applied to indoor scenarios, due to information loss from height compression[8]. Thus, a coarse-to-fine framework for the Gaussian representation can help delineate the overall structure at coarser levels and progressively focus on **finer details** at higher resolutions, enabling better modeling of complex and irregular shapes in indoor unstructured scenarios for occupancy prediction..
>
> ## **Q4 Novelty (Weakness 1)**
>
> From the provided citation[19,20], 3D AIGC focuses on generating rich meshes, spatially varying textures (e.g., textured Gaussians), and editable 3D content, which are not primary concerns for embodied scenarios. These methods mainly focus on directly 3D Gaussian rendering, not Gaussian-based 3D semantic occupancy prediction. Such methods  **cannot** be directly applied to this different type of task.
>
> Prior coarse-to-fine pipelines typically rely on convolution-based architectures[18]. In contrast, our work demonstrates that direct optimization in the Gaussian parameter space provides a more efficient and robust alternative for embodied occupancy prediction. This is particularly important in complex indoor environments where fine-grained, occlusion-aware reasoning is needed.
>
> To the best of our knowledge, no existing autoregressive pipeline operates directly on Gaussian representations for this task, primarily because conventional convolutional models are not well-suited to handle sparse and continuous Gaussian primitives. Our approach opens a new and promising direction for efficient, structured 3D scene understanding in embodied AI, particularly suited for modeling fine-grained geometric and semantic features in complex indoor environments.
>
> [1]Katyal, Kapil D., et al. "High-speed robot navigation using predicted occupancy maps." ICRA 2021.
>
> [2]Kim, Jinkyu, et al. "Stopnet: Scalable trajectory and occupancy prediction for urban autonomous driving." ICRA 2022
>
> [3]Asghar, Rabbia, et al. "Vehicle motion forecasting using prior information and semantic-assisted occupancy grid maps." IROS 2023.
>
> [4]Wang, Xiaofeng, et al. "Openoccupancy: A large scale benchmark for surrounding semantic occupancy perception." ICCV 2023
>
> [5]Ess, Andreas, et al. "Moving obstacle detection in highly dynamic scenes." ICRA 2009.
>
> [6]Tian, Xiaoyu, et al. "Occ3d: A large-scale 3d occupancy prediction benchmark for autonomous driving." NeurIPS 2023
>
> [7]Zheng, Yupeng, et al. "Monoocc: Digging into monocular semantic occupancy prediction." ICRA 2024.
>
> [8]Yu, Hongxiao, et al. "Monocular occupancy prediction for scalable indoor scenes." ECCV 2024.
>
> [9]Jiang, Ziyu, et al. "Peek-a-boo: Occlusion reasoning in indoor scenes with plane representations." CVPR 2020
>
> [10]Yuan, Xiaoding, et al. "Robust instance segmentation through reasoning about multi-object occlusion." CVPR 2021
>
> [11]Gan, Wanshui, et al. "Gaussianocc: Fully self-supervised and efficient 3d occupancy estimation with gaussian splatting." ICCV 25
>
> [12]Huang, Yuanhui, et al. "Gaussianformer: Scene as gaussians for vision-based 3d semantic occupancy prediction." ECCV 24
>
> [13]Paria, Biswajit, et al. "Minimizing flops to learn efficient sparse representations." ICLR 2020.
>
> [14]Wang, Jiabao, et al. "Opus: occupancy prediction using a sparse set." NeurIPS 2024
>
> [15]Liu, Haisong, et al. "Fully sparse 3d occupancy prediction." ECCV 2024
>
> [16]Lu, Yuhang, et al. "Octreeocc: Efficient and multi-granularity occupancy prediction using octree queries."  NeurIPS 2024
>
> [17]Wald, Johanna, et al. "Beyond controlled environments: 3d camera re-localization in changing indoor scenes." ECCV 2020
>
> [18]Wei, Yi, et al. "Surroundocc: Multi-camera 3d occupancy prediction for autonomous driving." ICCV 2023.
>
> [19]Chen et al., 3DTopia-XL: Scaling High-quality 3D Asset Generation via Primitive Diffusion, CVPR 2025.
>
> [20] Zhang et al., GeoLRM: Geometry-Aware Large Reconstruction Model for High-Quality 3D Gaussian Generation, NeurIPS 2024.

---

> > ### Comment · Reviewer_SDTU · 2025-08-01
> >
> > Thank you for your reply. We agree with Q2-4, but find the Q1 response unconvincing.
> >
> > 1. The authors claim 3DGS cannot provide compact, discrete, memory-efficient representations for robotics. However, 3DGS is already compact, memory-efficient, and uniform. It's unclear why robots need discrete representations when operating in continuous space. Recent work like GaussNav (TPAMI 2025) and GS-Planner (IROS 2024) successfully uses 3DGS for robotic navigation.
> >
> > 2. The cited literature [1-6] doesn't support the authors' claims. References [2-6] directly address autonomous driving, not indoor embodied scenes. Reference [1] only predicts 2D occupancy for small-scale driving scenarios.

---

> ### Author Response · Authors · 2025-08-02
> **Thank you very much for your thoughtful feedback!**
>
> Dear Reviewer SDTU:
>
> Thank you! We agree that the previous version did not sufficiently clarify the necessity of discretized representations in exploration-focused embodied scenarios. While autonomous driving is a subarea of robotics, additional examples specific to embodied agents are necessary, and we have now included them to strengthen our argument.
>
> Discretized representations are essential for mainstream **exploration** algorithms such as A* and *frontier detection*, which are **inherently** designed to operate over **discrete** graph[5,6] structures. The GaussNav(TPAMI25) mentioned from the feedback is a clear example, using **2D grid-based** *frontier detection*(Fig.3) for **exploration**(first episode, unseen) and refining its continuous representation during exploitation(following episodes). The GS-Planner(IROS24) applies a 3D voxel(**occupancy**) map(Fig2, unobserved regions and Section IV-B) to identify uncertain areas and guide exploration for continuous-space SLAM refinement. Additionally, its trajectory planning relies on MINCO, which is specifically designed for quadrotor (UAV) dynamics and is rarely applied in our ground robot scenarios. Furthermore, its experimental results(Tab1) **only** compare voxel raycasting(NeRF) with 3DGS in terms of **runtime**, without providing any quantitative evaluation of trajectory quality or reconstruction accuracy, nor demonstrating whether this UAV-based method is transferable to our ground-based exploration settings.
>
> Despite progress in outdoor 3D occupancy prediction, few methods suit indoor embodied agents requiring exploration, where research is still underdeveloped(EmbodiedOcc ICCV25). Typical use cases include online navigation[1,7], Occupancy Grid Mapping[2,3], and Online 3D Object Detection and Tracking in Dynamic Scenes to reduce search space[4]. Our method, mainly focused on exploration(e.g.,tab3 continuous-view, online setting) rather than exploitation, addresses the above challenges via an autoregressive framework that enables progressive spatial reasoning under partial observations, crucial for advancing online 3D understanding in indoor embodied settings.
>
> [1]Popović, Marija, et al. "Volumetric occupancy mapping with probabilistic depth completion for robotic navigation." IEEE Robotics and Automation Letters 6.3 (2021): 5072-5079.
>
> [2]Moravec, Hans, et al. "High resolution maps from wide angle sonar." ICRA 1985
>
> [3]Thrun, Sebastian. "Probabilistic robotics." Communications of the ACM 45.3 (2002): 52-57.
>
> [4]Sun, Jiaming, et al. "You don't only look once: Constructing spatial-temporal memory for integrated 3D object detection and tracking." ICCV 2021
>
> [5]Hart, Peter E., Nils J. Nilsson, and Bertram Raphael. "A formal basis for the heuristic determination of minimum cost paths." IEEE transactions on Systems Science and Cybernetics 4.2 (1968): 100-107.
>
> [6]Yamauchi, Brian. "A frontier-based approach for autonomous exploration." Proceedings 1997 IEEE International Symposium on Computational Intelligence in Robotics and Automation CIRA'97.'Towards New Computational Principles for Robotics and Automation'. IEEE, 1997.
>
> [7]Akmandor, Neşet Ünver, et al. "Deep reinforcement learning based robot navigation in dynamic environments using occupancy values of motion primitives." IROS 2022

---

> > ### Comment · Reviewer_SDTU · 2025-08-04
> >
> > Thank you for your reply! That is convincing. I've raised my rating to 4.

---

> > > ### Author Response · Authors · 2025-08-04
> > > **Thank you!**
> > >
> > > Dear Reviewer SDTU:
> > >
> > >
> > > We sincerely appreciate your thoughtful feedback during the first review round and are thankful for your improved rating following our rebuttal.
> > >
> > > Your input has been invaluable in strengthening our work, and we truly recognize the time and care you devoted to evaluating our submission.
> > >
> > > Thank you again!

---

### Official Review · Reviewer_pKS8 · 2025-07-01

**Clarity:** 2
**Significance:** 2
**Originality:** 3
**Rating:** 5
**Confidence:** 4

**Summary:**

This paper proposes a multi-scale Gaussian representation framework for 3D occupancy prediction. It proposes a multi-scale Gaussian encoding method to convert image features into hierarchical Gaussian primitives (position, scale, opacity, semantics). Then It employs an autoregressive refinement to progressively refine Gaussian parameters from coarse to fine scales via cross-attention. To reduce computation, it proposes a dynamic focused masking, which selectively updates informative Gaussians (top-ρ% by learned scores).
Evaluated on indoor (Occ-ScanNet, EmbodiedOcc) and outdoor (SSCBench-KITTI-360) benchmarks, DFGauss achieves SOTA results (e.g., 47.03 mIoU on Occ-ScanNet) with lower latency than voxel-based methods.

**Questions:**

1. In high-resolution settings, does the large number of Gaussian points lead to excessive memory consumption?

2. Does the method rely too heavily on the effectiveness of monocular depth estimation?

3. How robust is the model under large viewpoint changes?

**Ethical Concerns:**

["NO or VERY MINOR ethics concerns only"]

**Final Justification:**

In the rebuttal, the paper provides more ablations on the DepthAnything module. The designed Gaussian Encoder can avoid the metric ambiguity issues and boost the performance. In summary, the paper provides a novel method for the 3D occupancy prediction. It ablates all designs in detail. I recommend accepting this paper.

**Limitations:**

1. The visual comparisons and results cannot demonstrate the robustness and superiority of the method.

2. The paper does not discuss the effect of monocular depths.

**Paper Formatting Concerns:**

No formatting concerns.

**Quality:**

3

**Strengths And Weaknesses:**

Strengths:
1. The proposed multi-scale Gaussian pipeline has lower latency than existing methods.
2. In the benchmark, it performs better than existing SOTA methods.

Weakness:

1. Section 3.2 contains so many module design details, which are not clear.

2. The paper uses DepthanythingV2 to predict multi-scale depths for geometric alignment during Gaussian initialization. However, the depths from DepthAnythingV2 are metric-unknown and temporally inconsistent. It may also contain many artifacts. I believe this will affect the Gaussian initialization. However, the paper does not discuss this. Why not use other metric depth models? In outdoor scenes, such as KITTI, DepthAnythingV2 works worse in far regions. Will this affect the Gaussian?

3. The paper actually attempts to solve the occupancy prediction from a single view. It should compare with more occupancy methods.

4. The paper only shows 2 visual results in the paper. It is hard to evaluate the performance and quality of predicted occupancy. Furthermore, the paper argues that it targets autonomous navigation for an embodied agent. The temporal consistency is important for robotics navigation. The paper does not present temporal results and evaluation. The single-frame evaluation cannot support the application scenarios.

5. In indoor scenes, there are so many complex and fine-grained structures and objects. Could the method also work well on such objects?

---

> ### Author Rebuttal · Authors · 2025-07-30
>
> Dear Reviewer pKS8,
>
> Thank you for identifying the strengths of our work, including the low-latency design and competitive performance compared to existing methods. Below are our detailed responses to your feedback.
>
> ## **Q1 High-resolution settings**
>
> | Method                    | # Gaussians | Latency (ms) | Memory (MB) | Grids Size       |
> |---------------------------|-------------|--------------|-------------|------------------|
> | MonoScene (CVPR 2022)     | -           | 870          | 20723.2     | 256×256×32       |
> | ISO (ECCV 2024 )          | -           | 650          | 10214       | 60×60×36         |
> | EmbodiedOcc (ICCV 2025)   | 12600       | 114          | 3161        | 60×60×36         |
> | Ours                      | 12600       | 232          | 5732        | 60×60×36         |
> | Ours                      | 25600       | 265          | 6531        | 256×256×32       |
>
> Despite increasing both the grid size from 60×60×36 to 256×256×32 and the number of Gaussians from 12.6K to 25.6K, the memory usage only grows from 5732 MB to 6531 MB — an increase of approximately 13.9%. This demonstrates that the proposed hierarchical Gaussian refinement and selective masking mechanism scale efficiently, avoiding excessive memory consumption even under high-resolution settings.
>
> ## **Q2 Heavily rely on the monocular depth(weakness2)**
>
> We apologize for the lack of clarity. Due to space limitations, we briefly mention the existing depth estimation module to focus on presenting our primary contribution in detail. We adopt DepthAnythingV2 because it is **also** used in our baselines, EmbodiedOcc (ICCV 2025) and ISO (ECCV 2024), ensuring a fair comparison under the same monocular depth setup. However, our method does not directly rely on raw depth outputs. Instead, we adopt the Gaussian encoder design from [1,2], which includes an internal learnable refinement multi-layer MLP module to enhance the depth features. These refined features are then used to initialize Gaussian parameters, which are further processed through an image cross-attention module[2] and a feature refinement module[1]. In this way, depth is treated as a geometric prior, while the final occupancy prediction is driven by adaptive learning across scales.
>
> We also report results using depth predictions from DepthAnythingV2 directly, without the refinement module. As shown in the table below, using naive monocular depth features without further processing leads to a noticeable drop in performance. This highlights the importance of the refinement module in enhancing depth features and improving the overall occupancy prediction quality.
>
> #### *The results are presented in multiple tables due to Markdown formatting constraints that limit the width of single tables*
>
> | Method          | IoU   | mIoU  |
> |----------------|--------|--------|
> | Ours (refine)  | **39.89** | **14.58** |
> | Ours (naive)   | 34.26     | 11.95     |
>
> | Method         | car   | bicycle | motorcycle | truck | other-veh. | person | road  | parking |
> |----------------|-------|---------|------------|--------|-------------|--------|-------|---------|
> | Ours (refine)  | 22.21 | 1.85    | 4.88       | 14.78  | 5.97        | 2.03   | 54.23 | 15.78   |
> | Ours (naive)   | 18.56 | 0.21    | 1.92       | 13.26  | 3.32        | 0.35   | 52.22 | 13.21   |
>
> | Method         | sidewalk | other-grnd | building | fence | vegetation | terrain | pole | traf.-sign | other-struct. | other-object |
> |----------------|----------|-------------|----------|--------|------------|---------|------|-------------|----------------|---------------|
> | Ours (refine)  | 31.89    | 4.52        | 32.28    | 6.12   | 29.02      | 18.63   | 4.02 | 4.25        | 6.03           | 4.03          |
> | Ours (naive)   | 24.86    | 3.91        | 31.17    | 3.68   | 20.34      | 15.35   | 4.37 | 1.22        | 4.17           | 2.92          |
>
> ## **Q3 Large viewpoint results**
>
> The continuous-view experiment in Table 3 corresponds to a large-viewpoint setting, as it consists of 30 sequential frames with associated camera poses for each scene (line 464). This setup includes evaluating the union of frustums across 30 frames, thereby incorporating temporal consistency and naturally involving significant viewpoint variation and spatial transitions—making it a suitable benchmark for assessing robustness under large viewpoint changes.
>
> ## **Q4 Single-view occupancy prediction (weakness3)**
>
> We would like to clarify that our method is not limited to single-view prediction. We adopt the confidence-guided refinement strategy from[1](line 119), which enables temporal accumulation of information across frames. Our experiments in Table 3 demonstrate that DFGauss also improves performance in the continuous-view setting, where the model observes and integrates 30 sequential monocular frames with associated poses (line 464). This validates that our framework supports both single-view and continuous-view (embodied) occupancy prediction. Moreover, monocular occupancy prediction remains an important research problem due to its cost efficiency and relevance to real-world embodied agents, especially in camera-only systems[3,4]. Our contribution aims to advance this line of work by enhancing prediction quality in both single-frame and sequential-view settings.
>
> ## **Q5 Insufficient Visual and Temporal Evaluation(weakness4)**
>
> Thank you for the suggestion. Due to space limitations, we included only two qualitative examples in the main paper, but we provide additional visual results in the appendix. We will further expand this section in the camera-ready version to better illustrate the qualitative improvements of our method. While we cannot submit additional figures through the rebuttal system, we agree that visualizations are important for evaluating spatial and temporal consistency. Regarding temporal consistency, we note that Tables 3 and 4 present continuous-view results computed over the accumulated occupancy from 30 sequential monocular frames. These metrics **inherently** reflect the alignment and consistency of per-frame predictions over time. The observed improvements in global IoU and mIoU indicate that our model maintains robust predictions across diverse viewpoints, which is critical for navigation tasks in robotics.
>
> ## **Q6 Complex structures performance (weakness5)**
>
> As shown in Tables 1–4, DFGauss consistently improves the prediction of most object categories, particularly complex and fine-grained structures such as furniture and small objects. We observe that large planar regions like walls and ceilings show slightly lower improvements, which may indicate a trade-off: our dynamic refinement strategy tends to focus more on localized and semantically rich regions, potentially at the cost of global consistency in large homogeneous areas. This suggests an interesting direction for future work—developing strategies to balance detail-aware refinement with broader spatial regularization.
>
> [1] Wu, Yuqi, et al. "Embodiedocc: Embodied 3d occupancy prediction for vision-based online scene understanding." ICCV 2025
>
> [2] Huang, Yuanhui, et al. "Gaussianformer: Scene as gaussians for vision-based 3d semantic occupancy prediction." ECCV 2024
>
> [3]Yu, Hongxiao, et al. "Monocular occupancy prediction for scalable indoor scenes." ECCV 2024
>
> [4]Zheng, Yupeng, et al. "Monoocc: Digging into monocular semantic occupancy prediction." ICRA 2024

---

> > ### Comment · Reviewer_pKS8 · 2025-08-09
> > **comments**
> >
> > Dear authors,
> >
> > Thank you for your detailed replies. My main concern about the DepthAnything module has been solved. In the provided table, your refined results are much better than the naive one using depthanything.  I will raise the score to weak accept.

---

> > > ### Author Response · Authors · 2025-08-09
> > > **Thank you!**
> > >
> > > Dear Reviewer pKS8,
> > >
> > > We are deeply grateful for the time, care, and thoughtful feedback you have provided throughout the review process, and we sincerely value the effort you devoted to carefully evaluating our work, offering constructive insights, and helping us meaningfully strengthen our submission through your detailed and considerate assessment.
> > >
> > > Thank you once again for your time and support!

---

> ### Author Response · Authors · 2025-08-04
> **Thank You and Follow-Up on Review Feedback**
>
> Dear Reviewer pKS8,
>
> We sincerely thank you for your valuable feedback! With the discussion deadline approaching, we want to ensure that all of your concerns are fully addressed.
>
> If there are any additional clarifications or analyses you would like us to provide, please don’t hesitate to let us know.
>
> Thank you again, and best wishes!!!

---

> ### Author Response · Authors · 2025-08-08
> **Follow-Up on Review Feedback**
>
> Dear Reviewer pKS8,
>
> Thank you once again for your feedback on our paper and the time you’ve dedicated to reviewing it.
>
> We noticed that there is only **one day left** in the discussion period, and we would greatly appreciate any additional feedback you might have.
>
> Thank you again for your time and support in this process. Please don’t hesitate to reach out with any further questions or thoughts.
>
> Best regards,

---

### Official Review · Reviewer_txXB · 2025-07-01

**Clarity:** 2
**Significance:** 3
**Originality:** 3
**Rating:** 4
**Confidence:** 5

**Summary:**

This paper proposes a hierarchical Gaussian refinement strategy as an improvement over the conventional EmbodiedOcc framework. The proposed DFGauss mainly consists of a multi-scale Gaussian encoder, a hierarchical Gaussian refinement process, and a multi-scale supervision strategy. The paper conducts extensive experiments on both indoor and outdoor datasets, which demonstrates good performance compared with the baseline.

**Questions:**

1. How are the parameters used to compute the confidence score for Gaussians updated? Please refer to the weakness section for details.
2. How is the GPU memory consumption of DFGauss compared with the baseline method EmbodiedOcc?

**Ethical Concerns:**

["NO or VERY MINOR ethics concerns only"]

**Final Justification:**

Post rebuttal, my concerns regarding the method details and efficiency have been addressed. Thus I am leaning towards accepting the paper.

**Limitations:**

yes.

**Quality:**

3

**Strengths And Weaknesses:**

Strengths:
1. Good motivation. The multi-scale Gaussian encoder and hierarchical refinement design stem from the coarse-to-fine nature of visual perception.
2. Extensive experiments. The paper conducts experiments on two indoor datasets (excluding the mini versions) and one outdoor dataset to verify the effectiveness of the proposed method.

Weaknesses:
1. Potentially inappropriate expression. The "Dynamic Focused" expression in the title might be misleading because the paper actually focuses on static scenes and this phrase actually refers to adaptive selection of Gaussians requiring refinement. Also, the paper have potentially overclaimed itself to be autoregressive, I think "hierarchical" might be more suitable.
2. Technical flaw. The Gaussian refinement module first computes a confidence score and select Gaussians to refine with a percentile-based strategy. The computation of confidence scores should introduce learnable parameters, but the percentile-based selection is not differentiable which prevents gradients from back-propagating to the score-related parameters. The paper does not discuss this.

---

> ### Author Rebuttal · Authors · 2025-07-30
>
> Dear Reviewer txXB,
>
> Thank you for recognizing the strengths of our work, particularly the multi-scale Gaussian design and extensive experimental evaluation. Below are our detailed responses to your feedback.
>
> ## **Q1: Differentiable Scores (Weakness 2 flaw)**
>
> We appreciate the reviewer’s observation regarding the differentiability of the subset selection. In our implementation, we adopt the off-the-shelf, training-free differentiable top‑$k$ operator [1]. The $I_{\rho}$ in Equation (7) of the paper represents a **soft** index set corresponding to the top‑$\rho$% of Gaussians, enabling end-to-end differentiability. This choice allows gradient flow from the final loss back through the selection mechanism.
>
> After computing the per-Gaussian importance scores $\mathbf{q} \in \mathbb{R}^{B \times N}$ using a lightweight scoring MLP, we apply the differentiable top-*k* operator to produce a soft selection matrix  $I_{\text{select}} \in \mathbb{R}^{B \times N \times k}$ be a soft index set, where $B$ is the batch size, $N$ is the number of Gaussians, and $k$ is the top‑$\rho$% number of Gaussians. Each slice over the last dimension defines a distribution over $N$ elements, indicating their soft selection probability of being in the top‑$\rho$%.
> It identifies regions requiring further refinement, as $\mathbf{G}_{\text{select}} \in \mathbb{R}^{B \times K \times D}$ is selected from the original matrix $\mathcal{g} \in \mathbb{R}^{B \times N \times D}$ , where $D$ is the feature dimension.
>
> We apologize for the omission and will add this clarification to the appendix.
>
> ## **Q2 Memory consumption**
>
> | Method                    | Latency (ms) | Memory (MB) |
> |---------------------------|--------------|-------------|
> | ISO (ECCV 2024 )          | 650          | 10214       |
> | EmbodiedOcc (ICCV 2025)   | 114          | 3161        |
> | Ours                      | 232          | 5732        |
>
> As shown in the table, compared to traditional convolution-based architectures such as ISO and EmbodiedOcc, our method remains efficient for occupancy prediction, achieving a favorable balance between latency and memory consumption.
>
> ## **Q3 Potentially Inappropriate (Weaknesses 1)**
>
> We acknowledge the reviewer’s concern regarding the use of **Dynamic** in our title. Here, we clarify that “Dynamic” refers to the adaptive, content-aware computation mechanism in our model rather than implying temporal variation or scene motion. Our design aligns with the notion of **dynamic neural networks**, as characterized in [8], where “dynamic” refers to models whose computational graphs or activation patterns change depending on input instances—enabling efficient, selective processing. Our model exhibits the same key characteristics: (1) per-sample adaptive behavior, (2) selective computation for efficiency, and (3) sparse activation within the computational structure. Thus, our use of the term “Dynamic Focused” aims to describe the model architecture type, consistent with the terminology adopted in this literature, rather than the specific task or input modality.
>
> According to the formal definition of autoregressive models[2], such models forecast the variable of interest using a combination of its **previous** values. Our proposed model follows this principle by refining the current fine-grained Gaussian features based on **previously** generated coarse-grained outputs, which aligns with the autoregressive modeling paradigm. Although Visual Autoregressive (VAR) models [7] exemplify this idea in the 2D image domain by performing next-resolution prediction over discrete tokens or pixels, the autoregressive principle has also been adapted across various other domains, including normalizing flows [3], 3D object generation [4], monocular depth estimation [5], and offline policy learning [6]. In the same spirit, our work applies autoregressive refinement across spatial scales in the 3D Gaussian parameter space, forming a structured coarse-to-fine **autoregressive dependency** that is unique to our Gaussian-based 3D scene representation.
>
> [1] Petersen, Felix, et al. Differentiable Top-k Classification Learning. ICML, 2022.
>
> [2] Hyndman, Rob J., and George Athanasopoulos. *Forecasting: Principles and Practice*. OTexts, 2018. Section 8.3: Autoregressive models.
>
> [3] Bhattacharyya, Apratim, et al. "Normalizing flows with multi-scale autoregressive priors." CVPR 2020
>
> [4] Chen, Yongwei, et al. "SAR3D: Autoregressive 3D object generation and understanding via multi-scale 3D VQVAE." CVPR 2025
>
> [5] Wang, Jinhong, et al. "Scalable autoregressive monocular depth estimation." CVPR 2025
>
> [6] Zhang, Michael R., et al. "Autoregressive dynamics models for offline policy evaluation and optimization." ICLR 2021
>
> [7] Tian, Keyu, et al. "Visual autoregressive modeling: Scalable image generation via next-scale prediction." Neurips 2024
>
> [8]Dynamic Neural Networks: A Survey  IEEE TPAMI 2021

---

> > ### Comment · Reviewer_txXB · 2025-08-05
> >
> > Thanks for the thoughtful response. My concerns regarding the method details and efficiency have been addressed. I will raise my score.

---

> > > ### Author Response · Authors · 2025-08-05
> > > **Thank you for your feedback!**
> > >
> > > Dear Reviewer txXB,
> > >
> > > We deeply appreciate your constructive insights during the initial review phase and are grateful for your updated assessment after considering our rebuttal.
> > >
> > > Your feedback has played a significant role in improving our work, and we sincerely value the time and effort you dedicated to reviewing our submission.
> > >
> > > Thank you once again!

---

> ### Author Response · Authors · 2025-08-04
> **Thank You and Follow-Up on Review Feedback**
>
> Dear Reviewer txXB,
>
> We sincerely thank you for your valuable feedback! With the discussion deadline approaching, we want to ensure that all of your concerns are fully addressed.
>
> If there are any additional clarifications or analyses you would like us to provide, please don’t hesitate to let us know.
>
> Thank you again, and best wishes!!!

---

### Official Review · Reviewer_9Hse · 2025-07-03

**Clarity:** 2
**Significance:** 1
**Originality:** 2
**Rating:** 2
**Confidence:** 4

**Summary:**

This paper presents DFGauss, a 3D occupancy prediction framework from single indoor images. Instead of using computationally intensive voxel grids, DFGauss operates on sparse 3D Gaussian representations with multi-scale autoregressive modeling to improve the efficiency. Moreover, a dynamic focused masking mechanism that improves efficiency by selectively updating only the most informative Gaussians, concentrating computational effort on important regions. Experiments show that DFGauss achieves state-of-the-art performance on indoor 3D occupancy prediction benchmarks.

**Questions:**

1. How do you use multi-scale autoregressive in your work?
2. Could you provide the details about the Multi-Scale Gaussian Encoder and illustrate the network details?
3. What kind of network is the first row in Table. 5? Please describe the baseline.

**Ethical Concerns:**

["NO or VERY MINOR ethics concerns only"]

**Final Justification:**

The authors did not address my concerns about the relationship between the proposed work and the previous GaussianFormer and VAR. The authors believe the proposed method is neither a generative model nor an autoregressive method.

**Limitations:**

yes

**Paper Formatting Concerns:**

None.

**Quality:**

1

**Strengths And Weaknesses:**

Strength:
1. The performance promotion is impressive.
2. The proposed method is easy to understand.

Weaknesses:
1. The proposed Multi-Scale Gaussian Encoder should be introduced with more details, such as the number of layers of cross-attention and self-attention, and how to leverage the generated image features and depth map.
2. There are many typos and inconsistencies in the paper: q in Eq. 5 is not defined in the manuscript. The notions of the Gaussian set are not consistent in Eq. 1 and Eq. 7.
3. More visualizations should be provided, including comparisons on different datasets and abalations.
4. The proposed method appears to be merely a multi-scale version of GaussianFormer , and the reviewer does not find any connection between the proposed work and visual autoregressive models.

---

> ### Author Rebuttal · Authors · 2025-07-30
>
> Dear Reviewer 9Hse,
>
> Thank you for acknowledging the method’s improved performance and intuitiveness. Below are our detailed responses to your feedback.
>
> ## **Q1 Multi-scale autoregressive detail (weakness 4: the connection with visual autoregressive models)**
>
> According to the formal definition of autoregressive models [1], such models forecast the variable of interest using a combination of its **previous** values. Our proposed model follows this principle by refining the current fine-grained Gaussian parameters based on **previously** generated coarse-grained outputs, which aligns with the autoregressive modeling paradigm. This is formally expressed in Eq. (2) of the paper as $p\left(g^{(s)} \mid g^{(s-1)}, a^{(s)}\right) = \prod_i p\left(g^{(s)}_i \mid g^{(s-1)}_i, a^{(s)}_i\right)$ , where the prediction of each finer-scale Gaussian is conditioned on its corresponding coarser-scale Gaussian and original fine-grained features from the input. Thus, our framework adheres to the broader definition of autoregression by modeling each finer-scale output as dependent on previously generated coarser-scale values, thereby aligning with the autoregressive paradigm.
>
> While Visual Autoregressive (VAR) models [2] demonstrate autoregressive modeling in the 2D image domain via pixel- or token-level next-resolution prediction, other areas have similarly extended the autoregressive principle to their own tasks, including normalizing flows [3], 3D object generation [4], monocular depth estimation [5], and offline policy learning [6]. Our work does **not** directly apply 2D VAR to the 3D setting; instead, we adopt the underlying autoregressive principle and reformulate it for spatial scale refinement in the 3D Gaussian parameter space. This forms a structured coarse-to-fine dependency unique to our sparse 3D Gaussian representation.
>
> ## **Q2 Details on the Multi-Scale Gaussian Encoder architecture**
>
> As mentioned in line 114, our framework builds upon the baselines [7, 8]. We focus primarily on describing our novel design components. For a fair comparison, modules that are identical to those in previous works reuse the same structure, including the number of layers and parameter settings.
>
> **In detail, our encoder includes the following components:**
>
> **Same** as in [7,8], our encoder includes an **image cross-attention and refinement module** to integrate Gaussian features with image features extracted by the backbone [9]. It estimates per-pixel depth and combines it with image features via a depth-aware MLP to initialize structured 3D semantic Gaussians in the camera coordinate system. Each Gaussian is represented by a vector composed of mean $\mathbf{m} \in \mathbb{R}^3$ , scale $\mathbf{s} \in \mathbb{R}^3$ , rotation quaternion $\mathbf{r} \in \mathbb{R}^4$ , opacity $o \in \mathbb{R}$ , and semantic logits $\mathbf{c} \in \mathbb{R}^C$ . The interactions between image features and Gaussians, as well as those among Gaussians, are all performed in the camera coordinate system. The initial Gaussian parameters are then sent to the self-encoding module, which uses 3D sparse convolution (rather than deformable attention) to enable interaction among the Gaussians. These parameters are subsequently voxelized to convert the sparse Gaussians into a format compatible with sparse 3D convolution, producing a sparse 3D grid of features, each centered on a Gaussian.
>
> The **image cross-attention module** extracts visual information from multi-view images to enhance the 3D Gaussian queries. For each Gaussian, a set of reference points is sampled around its mean, with offsets determined by its covariance to reflect its spatial extent. These 3D reference points are projected into the image planes using the known camera intrinsics and extrinsics. The corresponding image features at these projected locations are then gathered via deformable attention[11]. The aggregated visual cues are used to update the high-dimensional Gaussian queries, which guide the subsequent refinement of the Gaussian properties.
>
> The **refinement module** updates each 3D Gaussian $\mathbf{G} = (\mathbf{m}, \mathbf{s}, \mathbf{r}, \mathbf{o}, \mathbf{c})$ using its corresponding high-dimensional query vector $\mathbf{Q} \in \mathbb{R}^m$ . Here, $\mathbf{m}$ is the 3D center, $\mathbf{s}$ is the scale (covariance), $\mathbf{r}$ is the rotation quaternion, $\mathbf{o}$ is the opacity, and $\mathbf{c}$ denotes the semantic class logits. After aggregating contextual information through the self-encoding and image cross-attention modules, the model predicts the update amounts $\Delta \mathbf{G} = (\Delta \mathbf{m}, \Delta \mathbf{s}, \Delta \mathbf{r}, \Delta \mathbf{o}, \Delta \mathbf{c})$ , which are used to refine the Gaussian properties as $\mathbf{G}_{\text{new}} = (\mathbf{m} + \Delta \mathbf{m},\; \mathbf{s} + \Delta \mathbf{s},\; \Delta \mathbf{r} \otimes \mathbf{r},\; \mathbf{o} + \Delta \mathbf{o},\; \mathbf{c} + \Delta \mathbf{c})$ . Here, $\otimes$ denotes a special quaternion composition operator for updating the rotation, enabling spatially coherent and stable refinement across blocks. These Gaussian parameters are subsequently passed into the refinement hierarchy described in Q1, where they are progressively optimized across scales via the autoregressive process.
>
> We will address this in the appendix.
>
> ## **Q3 Network in the first row of Table 5 ?**
>
> The first row corresponds to the standard GaussianFormer model[7,8]. It first extracts image features using an image backbone [9], and simultaneously predicts depth features from[10], which are used to initialize the 3D Gaussian parameters in the camera coordinate system. These Gaussians are then processed by the Gaussian encoder [7,8] and projected into voxel space via Gaussian-to-voxel splatting [8] to produce the final occupancy prediction. Our method extends this baseline by introducing a multi-scale autoregressive refinement framework that progressively updates the Gaussian parameters. As shown in Table 5, removing this autoregressive structure leads to a noticeable drop in performance, highlighting its contribution.
>
> ## **Q4 Inconsistent Gaussian set(weakness 2)**
>
> The two Gaussian sets are **different**, so they are intentionally defined with different meanings. The symbol in Equation 1 represents the entire set of Gaussians at a given scale, while the symbol in Equation 7 refers to the subset of selected Gaussians identified by the importance-based masking mechanism.
>
> ## **Q5 q in Eq. 5 is not defined(weakness 2)**
>
> As noted in line 170 of the paper, Eq. 5 involves **quaternion multiplication** and is written as $ \mathbf{q}_i^{(s)} = \text{Normalize}(\mathbf{q}_i^{(s-1)} \otimes \Delta \mathbf{q}_i^{(s)}) $, where $ \otimes $ denotes quaternion multiplication and $\text{Normalize}$ ensures the result is a unit quaternion. The symbol $ \mathbf{q}_i^{(s)} $ represents the rotation quaternion of the $i$-th Gaussian at scale $s$ , $ \mathbf{q}_i^{(s-1)} $ is the previous rotation quaternion at the coarser scale, and $ \Delta \mathbf{q}_i^{(s)} $ is a learned rotation quaternion update (a delta quaternion). We apologize for the oversight and will clarify this definition in the final version to avoid confusion.
>
>
> [1] Hyndman, Rob J., and George Athanasopoulos. *Forecasting: Principles and Practice*. OTexts, 2018. Section 8.3: Autoregressive models.
>
> [2] Tian, Keyu, et al. "Visual **autoregressive** modeling: Scalable image generation via **next-scale** prediction." Neurips 2024
>
> [3] Bhattacharyya, Apratim, et al. "Normalizing flows with **multi-scale autoregressive** priors." CVPR 2020
>
> [4] Chen, Yongwei, et al. "SAR3D: **Autoregressive** 3D object generation and understanding via **multi-scale** 3D VQVAE." CVPR 2025
>
> [5] Wang, Jinhong, et al. "Scalable autoregressive monocular depth estimation." CVPR 2025.
>
> [6] Zhang, Michael R., et al. "Autoregressive dynamics models for offline policy evaluation and optimization." ICLR 2021
>
> [7] Wu, Yuqi, et al. "Embodiedocc: Embodied 3d occupancy prediction for vision-based online scene understanding." ICCV 2025
>
> [8] Huang, Yuanhui, et al. "Gaussianformer: Scene as gaussians for vision-based 3d semantic occupancy prediction." ECCV 2024
>
> [9] Tan, Mingxing, and Quoc Le. "Efficientnet: Rethinking model scaling for convolutional neural networks." ICML 2019
>
> [10] Yang, Lihe, et al. "Depth anything v2." Neurips 2024
>
> [11] Zhu et al., Deformable DETR: Deformable Transformers for End-to-End Object Detection ICLR 2021

---

> ### Author Response · Authors · 2025-08-04
> **Thank You and Follow-Up on Review Feedback**
>
> Dear Reviewer 9Hse,
>
> We sincerely thank you for your valuable feedback! With the discussion deadline approaching, we want to ensure that all of your concerns are fully addressed.
>
> If there are any additional clarifications or analyses you would like us to provide, please don’t hesitate to let us know.
>
> Thank you again, and best wishes!!!

---

> > ### Comment · Reviewer_9Hse · 2025-08-07
> > **New Official Comment by Reviewer 9Hse**
> >
> > The reviewer still finds no correlation between the proposed method and the generative models, as the proposed network is just a multi-scale GaussianFormer. Besides, all multi-scale methods like CONet in OpenOccupancy and SurroundOcc use the multi-scale strategy and generate the next-scale occupancy prediction based on the last scale prediction and feature. As a result, the reviewer holds the opinion that the work lacks novelty.

---

> > > ### Author Response · Authors · 2025-08-08
> > > **Follow-Up on Review Feedback**
> > >
> > > Dear Reviewer 9Hse,
> > >
> > > Thank you for your feedback. We respectfully clarify that our model is not a generative model, which, according to the definition[5], generates new data based on learned patterns or distributions from training data. In contrast, our method autoregressively refines existing patterns using information from previous steps and does not generate new data. A multi-scale structure alone does **not** imply **dependency** between layers. In our case, we adopt a multi-scale autoregressive structure [3,4], where finer-level representations are explicitly conditioned on coarser ones[2]. Our formulation applies this principle to Gaussian representations, modeling their dependencies across scales. This aligns with the formal mathematical definition of autoregression [1], as expressed in Eq. (2) of the paper as $p\left(g^{(s)} \mid g^{(s-1)}, a^{(s)}\right) = \prod_i p\left(g^{(s)}_i \mid g^{(s-1)}_i, a^{(s)}_i\right)$.
> > >
> > > The papers mentioned in the above review comments  illustrate scenarios that highlight the motivation for our method. OpenOccupancy (ICCV 2023) refines voxel features by **directly splitting** coarse-grained cells, which works in discrete voxel space. However, in Gaussian feature space, it is hard to directly split a coarse Gaussian into multiple finer Gaussians. Each Gaussian is defined by position, covariance, and feature—not bounded cells—making such splitting ambiguous. This process instead requires a **learned** or stochastic refinement mechanism rather than a simple deterministic rule. SurroundOcc (ICCV 2023) operates in Euclidean voxel space and directly applies deconvolution to predict occupancy. In contrast, Gaussian features are **fundamentally different** operations like deconvolution are not directly applicable. For example, parameters such as quaternions **cannot** be aggregated through standard weighted convolution or deconvolution used in Euclidean space. Additionally, our method is based on single-view (monocular) prediction, which is distinctly different [8] from SurroundOcc and OpenOccupancy, whose multi-scale autoregressive structures are **inherently** designed for and dependent on multi-view input. Moreover, they target outdoor environments, which are structurally consistent and predictable[6], while our focus is on indoor scenes, which are more complex due to human interaction, irregular geometry, and illumination changes[7]. These fundamental differences in input setting and scene characteristics make our approach fundamentally different.
> > >
> > >
> > > Therefore, none of the above methods are applicable to our indoor monocular occupancy prediction scenario, as multi-scale autoregression cannot be directly applied in Gaussian space. As a result, our contribution to embodied monocular occupancy prediction is **unique**.
> > >
> > >
> > >
> > >
> > >
> > > [1] Hyndman, Rob J., and George Athanasopoulos. *Forecasting: Principles and Practice*. OTexts, 2018. Section 8.3: Autoregressive models.
> > >
> > > [2] Tian, Keyu, et al. "Visual **autoregressive** modeling: Scalable image generation via **next-scale** prediction." Neurips 2024
> > >
> > > [3] Bhattacharyya, Apratim, et al. "Normalizing flows with **multi-scale autoregressive** priors." CVPR 2020
> > >
> > > [4] Chen, Yongwei, et al. "SAR3D: **Autoregressive** 3D object generation and understanding via **multi-scale** 3D VQVAE." CVPR 2025
> > >
> > > [5]Mitchell, Tom M. "Generative and discriminative classifiers: Naive bayes and logistic regression." Machine learning 1.2 (2010): 1-17.  chatper 3
> > >
> > > [6]Tian, Xiaoyu, et al. "Occ3d: A large-scale 3d occupancy prediction benchmark for autonomous driving." NeurIPS 2023
> > >
> > > [7]Wald, Johanna, et al. "Beyond controlled environments: 3d camera re-localization in changing indoor scenes." ECCV 2020
> > >
> > > [8]Zheng, Yupeng, et al. "Monoocc: Digging into monocular semantic occupancy prediction." ICRA 2024.

---

> > > ### Author Response · Authors · 2025-08-08
> > > **Follow-Up on Review Feedback**
> > >
> > > Dear Reviewer 9Hse,
> > >
> > > Thank you once again for your feedback on our paper and the time you’ve dedicated to reviewing it.
> > >
> > > We noticed that there is only **one day left** in the discussion period, and we would greatly appreciate any additional feedback you might have.
> > >
> > > Thank you again for your time and support in this process. Please don’t hesitate to reach out with any further questions or thoughts.
> > >
> > > Best regards,

---

### Decision · Program_Chairs · 2025-09-17

**Decision:**

Accept (poster)

**Comment:**

This submission receives 2 borderline accepts, 1 accept and 1 reject. The concerns of reviewer pKS8, SDTU, and txXB have been addressed successfully in the rebuttal discussion period. In terms of reviewer 9Hse, the remaining concern is the relationship between the proposed work and the previous GaussianFormer and VAR, which I believe is not a major issue and can be clarified in the revision. Given the strong performance of the proposed method as shown in the manuscript and responses of authors w.r.t. reviewers' concerns, I recommend acceptance of this submission.

To improve submission quality, authors are required to include clarifications and additional ablation studies in the revision, including but not limted to:
1. the relationship between the proposed work and the previous GaussianFormer and VAR
2. modification of the title especially 'dynamic' and 'autoregressive' to avoid confusion
3. ablation study on grid resolution
4. details and clarification of module design